# PixelWorld: How Far Are We from Perceiving Everything as Pixels?

Zhiheng Lyu[1,2]   Xueguang Ma[1]   Wenhu Chen[1,2]
[1]University of Waterloo    [2]Vector Institute, Toronto
{z63lyu,x93ma,wenhuchen}@uwaterloo.ca

## Abstract

Recent agentic language models increasingly need to interact with real-world environments that contain tightly intertwined visual and textual information, often through raw camera pixels rather than separately processed images and tokenized text. This shift highlights the need for a *unified perception* paradigm. To investigate this idea, we explore **Perceive Everything as Pixels** (PEAP) and introduce PIXELWORLD, a benchmark that renders natural-language, tabular, mathematical, and diagrammatic inputs into a shared pixel space. Experiments across multiple benchmarks show that PEAP achieves comparable performance to token-based approaches on semantic understanding tasks, suggesting that vision transformers can partially capture global textual semantics without explicit tokenization. In contrast, reasoning-intensive tasks such as mathematics and code show notable performance degradation, although Chain-of-Thought prompting helps mitigate this gap by compensating for missing symbolic structure. We further find that when visual and textual information are closely integrated, representing everything as pixels simplifies preprocessing and avoids cross-modal misalignment. PIXELWORLD thus provides a systematic and practical framework for evaluating unified vision–language models and facilitates further exploration of pixel-based multimodal learning.

## 1 Introduction

In recent years, large vision–language models (L-VLMs) (Wang et al., 2024a; OpenAI, 2025; Gemini, 2024) have achieved remarkable progress across diverse real-world tasks. However, these models still rely on distinct processing pipelines for different modalities—treating images as pixels and text as discrete tokens. Such disjoint tokenization introduces a *representation mismatch* between vision and language, which hampers unified multimodal understanding and complicates system design. As recent works (Zheng et al., 2024; Koh et al., 2024; Tellex et al., 2020; Driess et al., 2023) push toward **agentic systems** capable of perceiving and acting in complex environments—from physical navigation (Elnoor et al., 2024) and travel booking (Chen et al., 2024a) to code repair on GitHub (Yang et al., 2024)—this mismatch becomes increasingly consequential. In such intertwined visual–textual settings, maintaining separate tokenization and perception modules not only incurs high preprocessing overhead (Xie et al., 2024; Koh et al., 2024) but also leads to information loss and layout inconsistencies (Dagan et al., 2024; Chai et al., 2024), ultimately limiting the scalability and robustness of multimodal agents.

To address these limitations, we explore a unified perception approach: **Perceive Everything as Pixels** (PEAP). Building on earlier efforts that considered pixel-based representations (Singh et al., 2024; Zhang et al., 2024), we systematically examine how representing both text and visual inputs uniformly in pixel space affects model behavior. In this paradigm, a vision–language model (VLM) jointly models multimodal inputs without requiring separate tokenization or modality-specific encoders. To better understand the benefits and challenges of this approach, we introduce PIXELWORLD, a comprehensive benchmark suite designed to evaluate how well existing VLMs perform under the PEAP setting.

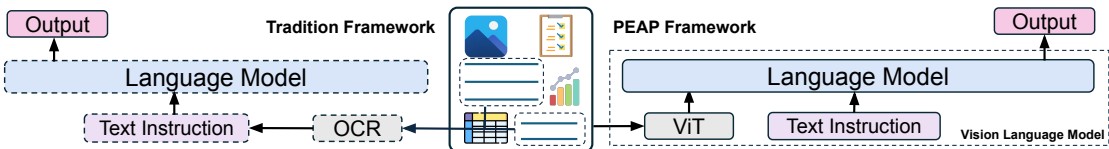

Figure 1: **Overview of the PEAP Framework.** PEAP (*Perceive Everything as Pixels*) unifies text, structural, and visual inputs into a single pixel space, where a Vision Transformer (ViT) encodes the pixels and a language decoder performs reasoning. Both components are enclosed within the dashed box to indicate that they jointly form a vision–language model (VLM). By eliminating modality-specific preprocessing such as OCR and tokenization, PEAP better aligns with human perception and reduces cross-modal engineering overhead.

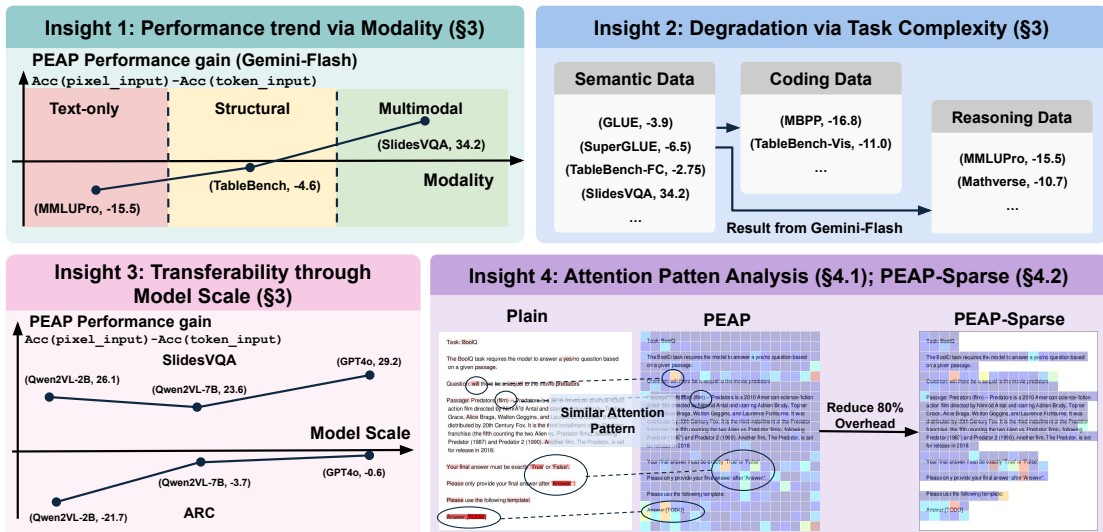

Figure 2: **Key Findings on the PixelWorld Benchmark.** Evaluated across text-only, structural, and multimodal settings (§2, §3), PEAP shows four major insights: (1) *Modality Trend:* consistent gains on layout-heavy and multimodal tasks such as websites, slides, and documents; (2) *Task Complexity:* performance degradation on reasoning- and code-centric benchmarks (see §3.1–§3.2); (3) *Transferability by Scale:* larger VLMs (e.g., GPT-4o, Gemini-Flash) exhibit smaller pixel–token gaps; and (4) *Attention and Efficiency:* text and image inputs show similar global attention patterns, while the proposed PEAP-Fast reduces up to 80% of computation overhead (§4.2).

In PixelWorld, we select 10 widely used benchmarks covering a diverse range of modalities and task scenarios. For each dataset, we construct both traditional token-based and pixel-based (PEAP) input formats using image synthesis and OCR techniques (see Table 1). We then evaluate vision–language models of varying scales, from Qwen2VL-2B to GPT-4o. Cross-modal evaluation in Section 3 reveals three overarching insights: **Insight 1**: In intrinsically multimodal settings such as website rendering, slide comprehension, and document understanding, PEAP consistently mitigates OCR noise and yields stronger performance; **Insight 2**: For reasoning-intensive tasks such as mathematics and code, pixelization leads to noticeable accuracy drops, though the gap narrows as model capacity increases—suggesting that scale plays a key role in enabling cross-modal transfer; and **Insight 3**: Larger models exhibit more robust instruction-following and long-context reasoning across modalities, whereas smaller models struggle, highlighting the importance of scale-aware training under the pixel-based paradigm.

To further interpret these findings, we conduct three complementary analyses. (1) Representation analysis: We visualize the attention patterns of Qwen2VL-7B and observe broadly similar global structures between

| Dataset Name | Size | Task | Modality Transfer | Split |
|---|---|---|---|---|
| **Text-only** | | | | |
| GLUE (Wang, 2018) | 59,879 | Natural language understanding | Synthesis | test |
| SuperGLUE (Sarlin et al., 2020) | 19,294 | Natural language understanding | Synthesis | test |
| MMLU-Pro (Wang et al., 2024b) | 12,032 | Domain knowledge and reasoning | Synthesis | test |
| ARC (Clark et al., 2018) | 3,548 | Science question answering | Synthesis | test |
| GSM8K (Cobbe et al., 2021) | 1,319 | Math problem solving | Synthesis | test |
| MBPP (Austin et al., 2021) | 757 | Programming tasks | Synthesis | test |
| **Structured** | | | | |
| TableBench (Wu et al., 2024) | 888 | Table data understanding and analysis | Synthesis | test |
| **Multimodal** | | | | |
| MathVerse (Zhang et al., 2025) | 788 | Math and visual reasoning | Natural | test |
| MMMU-Pro (Yue et al., 2024) | 1,730 | Multimodal reasoning | Synthesis | test |
| SlidesVQA (Tanaka et al., 2023) | 2,136 | Multimodal question answering | OCR | test |
| Wiki-SS (Ma et al., 2024) | 3,000 | Multimodal retrieval question answering | OCR | train |

Table 1: Overview of datasets categorized by modality, usage, size, and split. Modality Transfer means the method to adopt the dataset into counterpart modality. For OCR, we adopt the result from the origin datasets. For WikiSS-QA, since the positive document of the test set is not released, we subsample 3,000 training data points randomly to evaluate.

token- and pixel-based inputs, suggesting that certain aspects of language modeling behavior may transfer into the visual space, though not implying full equivalence. (2) Efficiency optimization: We measure inference latency and find that while PEAP increases computational cost due to larger input resolution, our proposed PEAP-Fast algorithm effectively prunes blank patches, achieving up to 80% speedup with negligible accuracy degradation. (3) Prompt sensitivity: We study prompting strategies and find that Chain-of-Thought (CoT) reasoning yields more consistent gains under the pixel-based representation compared to standard direct prompting, indicating potential synergies between reasoning supervision and visual encoding.

In summary, our contributions are as follows:

1. **PixelWorld Benchmark**: We present a unified benchmark that transforms text, structural, and multimodal datasets into pixel space, offering a direct and reproducible framework to evaluate the trade-offs between pixel- and token-based modeling. The benchmark and code are publicly released to facilitate standardized comparison and future research on multimodal perception.
2. **Task–scale insights**: Through large-scale evaluation, we show that PEAP improves layout-heavy or intrinsically multimodal tasks (e.g., website and document understanding) while reducing accuracy on reasoning- or code-centric tasks. The performance gap consistently narrows with model scale, underscoring the role of capacity in enabling cross-modal transfer.
3. **Efficiency & interpretability**: We propose PEAP-Fast, an inference-time pruning strategy that removes blank pixel patches, achieving up to a $3\times$ latency reduction with minimal loss in accuracy. Attention visualizations reveal partially shared global structures across modalities, providing an interpretable perspective on how visual encoders approximate token-level reasoning behavior.

## 2 Datasets

Several representative datasets covering different skill domains are selected, as shown in Table 1. We primarily utilize the prompts provided by the datasets. If no prompts are available, we apply a default prompt. By default, we employ Direct Prompting; however, for more complex and mathematical datasets such as MBPP (Austin et al., 2021), MMLU-Pro (Wang et al., 2024b), and MathVerse (Zhang et al., 2025), we adopt Chain-of-Thought (CoT) prompting to enhance performance. All evaluations are conducted in a zero-shot

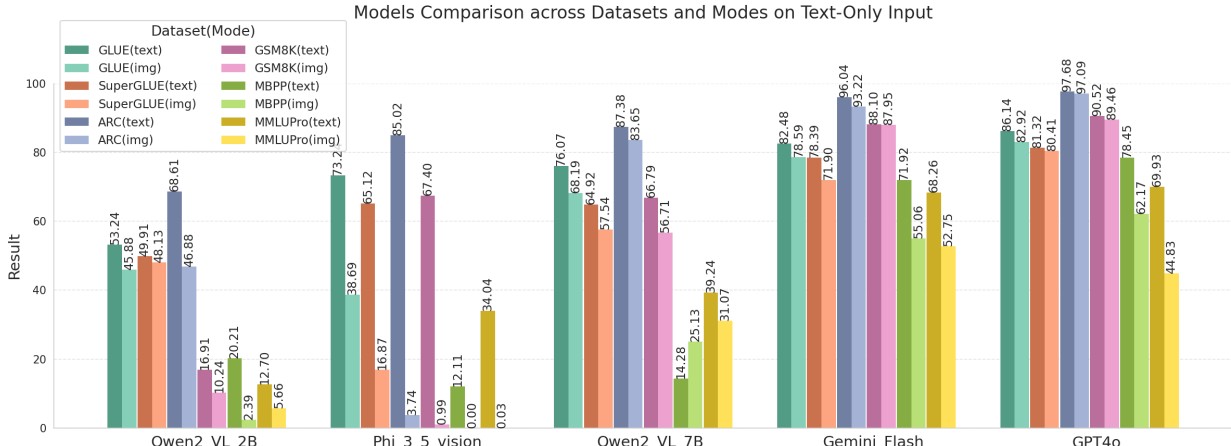

Figure 3: The performance of **text-only** datasets. The comparison is made between text input and synthesized image input. Most models demonstrate comparable performance on language understanding datasets such as SuperGLUE, GLUE, and ARC. However, notable performance disparities emerge between text-based input and synthesized image input on mathematical reasoning tasks (e.g., MMLU-Pro, GSM8K) and programming tasks (e.g., MBPP). Phi-3.5-Vision exhibits consistently poor performance across all vision tasks, primarily due to its insufficient instruction-following capabilities.

manner to mitigate potential performance degradation caused by the sensitivity of instruction-tuned large models to few-shot prompting.

To evaluate both Token-based and Pixel-based methods, we require paired Text-input and Image-input prompts. We adopted modality transfer strategies to reduce reliance on the information modality provided by existing datasets, as detailed in Table 1. For datasets categorized as *Text-Only* and *Structured*, all data is originally in plain text format, necessitating image synthesis prior to evaluation. For *Multimodal* datasets, textual content embedded in images is extracted using OCR, or the textual components provided by the original datasets are directly utilized for evaluation. Notably, the MathVerse dataset (Zhang et al., 2025) inherently includes a Text-Only modality, offering detailed textual descriptions of image-based information.

**Image Data Synthesis** For text-only and structured datasets, we developed an image data synthesis pipeline to generate diverse image inputs for evaluation. Image widths were adaptively adjusted between 512 and 1024 pixels based on text length, with a fixed height of 256 pixels. Font sizes ranged from 15 to 25 points, and padding varied from 5 to 30 pixels. To enhance robustness, we applied various types of noise, including radial, horizontal, vertical, and Multi-Gaussian noise, as well as high-frequency Gaussian noise to simulate distortions commonly introduced by real-world cameras. For structured datasets, such as tables, data was rendered as images using the Python package `dataframe_image`. Example inputs from different tasks are provided in Appendix A.

## 3 Experiments

In this section, we will detail our baseline, metrics and models. The experimental results will be organized by 'Text Input', 'Structued Input' and 'Multimodal Input'.

**Baseline** We establish the baseline by using the same VLMs with text-only prompts. To ensure fairness, we employ identical prompts and add the instruction *"Please follow the instruction in the image"* when applying PEAP. This ensures that the VLMs can correctly process instructions embedded within images. Ideally, the baseline and PEAP should yield equivalent performance. This comparison helps identify areas for improvement in existing VLMs.

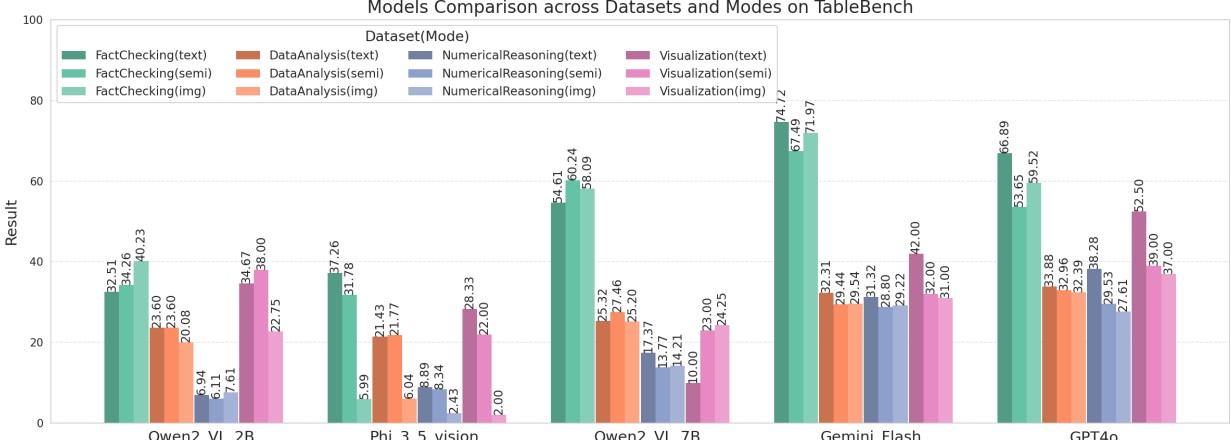

Figure 4: The performance of the **structured** dataset. We report all the subsets for the TableBench. In the *semi* setting, questions were presented as text, while tables were rendered as synthetic images. We observed that for tasks involving reasoning (numerical reasoning) and coding (visualization subset), synthetic images yielded inferior performance compared to text. However, for tasks emphasizing semantic understanding, such as data analysis and fact checking, synthetic images achieved performance comparable to or even surpassing text. Additionally, we found that the semi approach often performed worse than either text or synthetic images individually, providing insights into potential limitations and future directions for leveraging vision-language models (VLMs).

**Metrics** For QA tasks (*WikiSS-QA*, *SlidesVQA*, *TableBench*), we use *ROUGE-L*, which measures the longest common subsequence between prediction and reference to approximate answer overlap. We choose it for convenience and comparability, and expect other semantic metrics (e.g., BERTScore, LLM judges) to show similar trends. For classification benchmarks, including *MMLU-Pro*, *GLUE*, *SuperGLUE*, *ARC*, and *Math-Verse*, we use accuracy, which directly reflects the model's performance in selecting correct options. For *GLUE* and *SuperGLUE*, we follow their standard evaluation protocols, utilizing task-specific metrics such as Matthews correlation, F1 score, and Pearson correlation. For the code generation task *MBPP*, we evaluate performance using the pass@1 rate, which measures whether the generated code successfully passes all test cases. For the mathematical reasoning dataset *GSM8K*, we employ exact match accuracy, as these problems require precise numerical answers. For the visualization subtask of *TableBench*, following the original codebase, we treat it as a code generation task and evaluate the correctness of the generated visualizations.

**Model Selection** To validate PIXELWORLD, we selected a diverse set of vision-language models (VLMs) with varying scales to ensure the robustness and generalizability of our findings. It also allowed us to analyze the behavior of models across different sizes. We evaluated several widely used vision-language models (VLMs), including Qwen2VL-2B (Wang et al., 2024a), Phi-3.5-3.2B (Abdin et al., 2024), Qwen2VL-7B (Wang et al., 2024a), Gemini-Flash (Gemini, 2024)[1], and GPT-4o (OpenAI, 2025).

## 3.1 Text Input

Figure 3 reports model accuracy on text-only datasets (e.g., ARC, MMLU-Pro, GLUE, GSM8K, SuperGLUE, MBPP). Two major insights emerge:

**Better Transferability in Larger Models** Larger language models (e.g., GPT-4o, Gemini-Flash) exhibit better transferability between text and image-based performance, while smaller models struggle with both transferability and instruction following. For instance, on the ARC dataset, GPT-4o's performance declines by only 0.59 points when transitioning from text to synthetic images, whereas the smaller Qwen2-VL-2B suffers a substantial 21.73-point drop (from approximately 68.61 to 46.88). This trend suggests that more

---

[1]We use `gemini-1.5-flash-002`, which was the latest available version during this study.

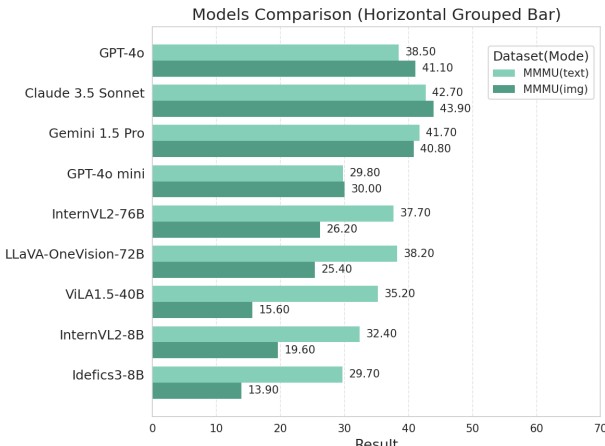

Figure 5: The performance of the **multimodal** dataset (MMMU-Pro). We adopt the result reported by the origin paper. We can observe that strong models perform better in PEAP.

capable models preserve their reasoning abilities across modalities, while smaller models face greater difficulty. Additionally, smaller models (e.g., Phi-3.5-vision) not only show weaker overall performance on standard benchmarks but also struggle significantly when instructions are presented as images. Their performance consistently lags behind that of larger models, particularly on tasks like MBPP. This supports *Insight 3* in Figure 2.

**Performance Degradation with More Complex Tasks** We observe significant drops on benchmarks requiring advanced reasoning, such as mathematical, coding or domain-specific tasks. For example, when moving from text to image inputs on the MMLU-Pro dataset, GPT-4o exhibits a drop of more than 25 points. In contrast, on GLUE and SuperGLUE, the decline remains under 5 points. These findings indicate that while existing large models achieve comparable performance between text and visual modalities on simpler tasks, a gap still exists at a deeper level in visual-based and text-based understanding, demonstrating room for improvement in modality adaptation training.

## 3.2 Structured Input

Figure 4 summarizes model performance on four TableBench subsets: Fact Checking, Data Analysis, Numerical Reasoning, and Visualization.

**Reasoning Complexity Impacts Performance** Fact Checking and Data Analysis show moderate performance drops, as they rely on semantic understanding. In contrast, Numerical Reasoning and Visualization—requiring more intricate reasoning and coding—exhibit larger declines when switching to synthetic images. Combined with *"Performance Degradation with More Complex Tasks"* in Section 3.1, this supports *Insight 2* in Figure 2.

**Smaller Performance Gaps with Structured Data** Compared to text-only tasks, structured tasks show smaller performance gaps between text and image inputs. Notably, Qwen2VL-2B even outperforms its text-based results on Fact Checking, suggesting robust visual representations can aid semantic tasks in smaller models.

**Challenges with Mixed-Modality Inputs** The "semi" format—where tables appear as images while questions remain text-based—performs worse than either fully text-based or fully image-based formats. This suggests that conventional VQA approaches, which process text and images using separate encoders, may be more susceptible to performance bottlenecks. As multimodal scenarios become increasingly prevalent, PEAP is expected to demonstrate superior performance compared to mixed-modality methods.

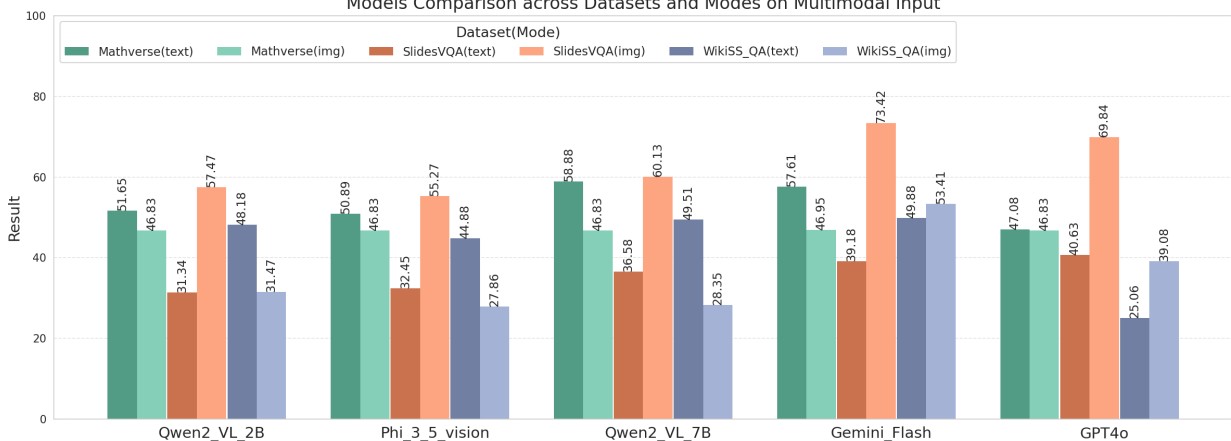

Figure 6: The performance of the **multimodal** datasets (except MMMU-Pro). We compare text-only and vision-only subsets in Mathverse, while SlidesVQA and WikiSS-QA are evaluated as VQA tasks. Larger models perform better on text-based tasks with more modalities. GPT-4o tends to generate longer responses in long-context QA, leading to performance degradation on WikiSS-QA.

## 3.3 Multimodal Input

Figure 6 presents model performance on multimodal datasets, including text-only and vision-only subsets of Mathverse and VQA tasks like SlidesVQA and WikiSS-QA. Results on MMMU-Pro (Figure 5) use reported values from the original paper. Three key observations emerge:

**Image Inputs Enhance Disambiguation** Incorporating images improves performance by reducing ambiguity compared to text-only benchmarks. In SlidesVQA, all models outperform their text-only baselines, while in WikiSS-QA and MMLU-Pro, visual context provides clarifying information, leading to accuracy gains in larger models. Combined with *"Smaller Performance Gaps with Structured Data"* in Section 3.2, this supports *Insight 1* in Figure 2.

**Challenges in Complex Reasoning** While multimodal inputs aid basic tasks, complex reasoning remains a bottleneck. In Mathverse, visual cues help but fail to support multi-step logical deductions. Even Gemini-Flash shows accuracy drops on intricate reasoning tasks. Additionally, WikiSS-QA poses challenges due to its long-context nature. Smaller models struggle with PEAP, and GPT-4o underperforms in token-based tasks, highlighting difficulties in processing extended contextual dependencies. This aligns with Sections 3.1 and 3.2.

**Larger Models Benefit More from Multimodal Data** Larger models gain more from multimodal inputs. On SlidesVQA, Gemini_Flash improves by 34.24 points, compared to Qwen2-VL-7B's 23.55-point boost. This suggests that larger models, with more extensive prior knowledge and advanced architectures, leverage multimodal data more effectively than smaller models.

# 4 Discussion

## 4.1 Q1: Does PEAP have the same attention?

To examine whether VLMs attend to similar regions when processing textual and image inputs, we visualize the average attention map of the final layer in Qwen2-VL-7B using a heatmap (Figure 7). Specifically, we analyze the model's behavior on a *BoolQ* example from SuperGLUE, comparing its attention patterns under text-based and image-based inference. Similar attention behaviors are observed across different datasets; more examples are shown in Appendix C.

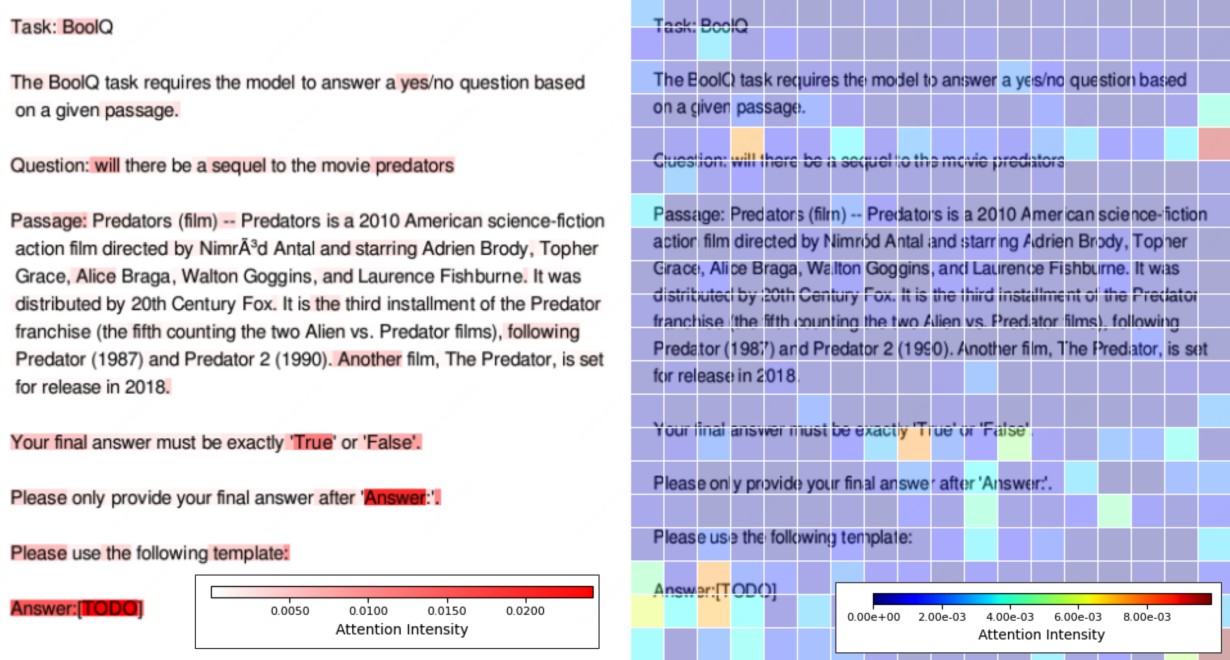

Figure 7: Last Layer Attention Heatmap on Qwen2VL-7B between token-based (left) and pixel-based (right) inference. Although the overall attention intensity on image inputs is generally lower, both modalities exhibit highly similar attention patterns.

**Computation of Heatmaps.** The attention heatmaps are computed during greedy decoding by averaging the last-layer attention across tokens. For multi-head attention, we apply a simple mean across heads. Formally, the attention weight for position $i$ is given by:

$$\text{Heatmap}(i) = \frac{1}{H} \sum_{h=1}^{H} \left| A_h^{(L)}[t, i] \right|, \quad i \in [s, e)$$

where $H$ denotes the number of attention heads, $L$ is the index of the last layer, $A_h^{(L)}[t, i]$ represents the attention weight from token $t$ to token $i$ in head $h$, and $[s, e)$ corresponds to the decoding range during greedy decoding.

**Observations.** As shown in Figure 7, Qwen2-VL-7B consistently focuses on task-relevant elements such as the question prompt ("will there be a sequel ..."), salient passage keywords (e.g., "film", "starring", "Alice"), and the expected answer format ("Answer: True/False"). This pattern remains stable across both textual and visual representations, suggesting that the model exhibits largely comparable attention behavior regardless of input modality. However, we also observe that certain blank patches in the image-based inputs occasionally receive disproportionately high attention weights, indicating that while the visual encoder aligns closely with the text encoder, it still introduces redundant activations.

### 4.2 Q2: How to make PEAP more efficient?

As a trade-off for generalization, image-based inference often requires significantly more computational resources than text-based inference. This is partly due to the additional overhead from the ViT backbone and higher redundancy in image tokens. To estimate the performance gap quantitatively, we conducted experiments on SuperGLUE (Table 2). The results show that inference latency for image-based inputs can exceed text-based methods by 150% to 250%.

| | SuperGLUE Evaluation Results | | |
|---|---|---|---|
| **Task** | **Text** | **PEAP** | **PEAP-Fast** |
| BoolQ | 79.69% | 82.11% | 80.89% |
| CB | 67.70% | 40.77% | 39.57% |
| COPA | 93.00% | 91.00% | 86.00% |
| MultiRC | 65.90% | 61.28% | 60.80% |
| ReCoRD | 12.54% | 5.94% | 6.08% |
| RTE | 82.31% | 72.92% | 77.26% |
| WiC | 53.29% | 55.80% | 55.64% |
| WSC | 63.46% | 65.38% | 59.62% |
| **Final Score** | 64.74% | 59.40% | 58.23% |

Table 2: Performance of *Qwen2VL-7B* on SuperGLUE dataset by Text, PEAP and PEAP-Fast. We can observe the comparable performance between PEAP and PEAP-Fast.

| | Inference Time (s) | | | Overhead (%) | |
|---|---|---|---|---|---|
| **Subset** | **Text** | **PEAP** | **PEAP-Fast** | **PEAP** | **PEAP-Fast** |
| BoolQ | 369 | 1,381 | 906 | 274.80 | 145.55 |
| CB | 8 | 22 | 15 | 175.00 | 87.50 |
| COPA | 39 | 38 | 22 | -2.56 | -43.59 |
| MultiRC | 609 | 3,861 | 2,550 | 534.80 | 318.71 |
| ReCoRD | 7,016 | 19,012 | 14,288 | 171.01 | 103.72 |
| RTE | 68 | 117 | 92 | 72.06 | 35.29 |
| WiC | 69 | 224 | 157 | 224.64 | 127.54 |
| WSC | 11 | 36 | 27 | 227.27 | 145.45 |
| **Total** | 8,089 | 24,690 | 18,051 | 205.27 | 123.19 |

Table 3: Inference Time (s) of *Qwen2VL-7B* on SuperGLUE dataset with single A100 server by PEAP and PEAP-Fast. We can observe a 82.08% overhead reduce on PEAP-Fast method. Overhead is calculated as the percentage increase in time relative to the text method.

To reduce redundancy in visual inputs, we propose **PEAP-Fast**, which first identifies empty patches via a simple variance-based threshold—if the pixel-value variance in a patch is lower than a preset threshold, that patch is treated as empty and is pruned from all attention computations. Crucially, we preserve the original positional embeddings for the remaining tokens, ensuring no loss of spatial layout perception. This strategy aligns with how humans naturally focus on salient regions rather than blank spaces, thereby significantly reducing context length without sacrificing structural information. Testing PEAP-Fast on SuperGLUE reveals a minor accuracy drop of only 1.17% (Table 2). More importantly, the average overhead decreases from 205.27% to 123.19%, yielding an 82.98% reduction (Table 3). These results demonstrate that removing empty patches offers substantial computational savings while maintaining strong performance, making image-based inference more practical for real-world deployments. Attention heatmap between PEAP and PEAP-Fast are shown in Appendix B.

### 4.3  Q3: Is PEAP sensitive to the prompting method?

In Section 3, results on purely textual synthesis tasks show that image-based inputs consistently underperform text inputs, likely due to dataset domain gaps and weaker instruction following on image representations. To address this, we applied CoT-style prompts to the SuperGLUE dataset to enhance cross-domain instruction following (Table 4). Notably, Qwen2VL-7B showed significant improvements in tasks where image input underperformed compared to text input, such as CB and RTE. Overall, CoT prompts improved image input performance by 2.58%, surpassing the 0.3% improvement observed for text input.

| Metric | Direct | | CoT | | Improve (CoT - Direct) | |
|---|---|---|---|---|---|---|
| | Text | PEAP | Text | PEAP | Text | PEAP |
| BoolQ | 79.88% | 81.71% | 81.13% | 80.73% | 1.25% | -0.98% |
| CB | 67.70% | 34.78% | 81.04% | 59.57% | 13.34% | 24.79% |
| COPA | 93.00% | 87.00% | 89.00% | 83.00% | -4.00% | -4.00% |
| MultiRC | 65.73% | 62.28% | 69.08% | 60.41% | 3.35% | -1.87% |
| ReCoRD | 12.50% | 5.88% | 6.37% | 4.66% | -6.13% | -1.22% |
| RTE | 82.31% | 72.92% | 83.03% | 77.26% | 0.72% | 4.34% |
| WiC | 52.82% | 54.39% | 54.39% | 53.92% | 1.57% | -0.47% |
| WSC | 65.38% | 61.54% | 57.69% | 61.54% | -7.69% | 0.00% |
| Overall | 64.92% | 57.56% | 65.22% | 60.14% | 0.30% | 2.58% |

Table 4: Comparison of Direct and CoT performance across Text and Image modalities, along with their respective improvements (CoT - Direct), presented as percentages.

## 5 Related Work

**Multimodal Large Language Models and Benchmarks** Recent progress in multimodal AI has led to the development of models like GPT-4o (OpenAI, 2025), Gemini (Gemini, 2024), and Claude-3.5 (Anthropic, 2025), which integrate vision-based training to improve instruction-following capabilities. Benchmarks for these models have evolved from task-specific datasets, such as VQA (Agrawal et al., 2016) and DocVQA (Mathew et al., 2021), to more comprehensive evaluations, including MMMU-Pro (Yue et al., 2024), MM-Bench (Liu et al., 2024), and MegaBench (Chen et al., 2024b). However, most current research focuses on the semantic understanding of visual content, with only a few benchmarks—such as MathVerse (Zhang et al., 2025) and MMMU-Pro (Yue et al., 2024)—addressing text recognition and comprehension within images. Our work shifts the focus towards evaluating how well large language models understand language through visual input compared to traditional token-based input.

**Screenshot LMs** Recent studies have demonstrated that pretraining on synthetic screenshots can enable vision-language models (VLMs) to achieve performance comparable to that of BERT on language modeling tasks (Lee et al., 2022; Rust et al., 2023; Gao et al., 2024). This approach allows models to better capture text structures without relying on OCR-based methods. Furthermore, our analysis highlights a performance gap between existing VLMs on vision-based tasks and their text-only counterparts, particularly in the absence of relevant pretraining. Interestingly, in certain scenarios, VLMs perform as well as or even better than text-only models, underscoring the potential of this research direction. In the context of document retrieval, recent advancements (Faysse et al., 2024; Ma et al., 2024) have shown that large-scale pretraining on screenshots can outperform traditional OCR-based methods, further reinforcing the advantages of vision-language pretraining.

**Language Tokenization** Tokenization methods, such as Byte Pair Encoding (BPE) (Shibata et al., 1999; Sennrich et al., 2016), are widely used in language modeling, but recent studies suggest that they may not always be optimal. For instance, MegaByte (Yu et al., 2023) demonstrated that fixed-length tokenization can improve both computational efficiency and cross-modal capabilities. Similarly, BLT (Pagnoni et al., 2024) proposed entropy-based tokenization, while LCM (team et al., 2024) emphasized the benefits of processing higher-level semantic concepts rather than individual tokens. Inspired by these approaches, we explore whether adaptive image patches can effectively infer textual meaning. At a higher level, we investigate the unification of text and image inputs into a shared representation space, enabling reasoning through abstract semantic concepts rather than traditional token-based methods.

## 6 Conclusion

We present PIXELWORLD, a benchmark that renders text, tables, code, and images as pixels, enabling direct evaluation of the *Perceive Everything as Pixels* (PEAP) paradigm. Experiments yield three main takeaways. **(1) Semantic understanding:** PEAP performs on par with token-based baselines on sentence- and paragraph-level tasks, and its patch-level attention exhibits similar global structures to token attention, suggesting partial transfer of language modeling behavior into the visual domain. **(2) Reasoning:** Performance drops on math, logic, and program-repair benchmarks, though Chain-of-Thought prompting mitigates but does not close this gap, highlighting the continued need for explicit reasoning structure. **(3) Multimodal perception:** Pixel-based inputs outperform OCR pipelines on websites, slides, and documents by preserving spatial context and avoiding recognition noise. To alleviate the higher computational cost of pixel inputs, we propose PEAP-Fast, which prunes blank patches and achieves up to a $3\times$ inference speedup with minimal accuracy loss. Overall, these findings illustrate both the potential and limitations of PEAP, positioning PIXELWORLD as a diagnostic and reproducible benchmark for studying unified vision–language representations and guiding future work on efficiency and reasoning in multimodal agents.

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

Figure 8: An example input of GSM8K dataset, using Direct Prompt.

Lili Yu, Dániel Simig, Colin Flaherty, Armen Aghajanyan, Luke Zettlemoyer, and Mike Lewis. Megabyte: Predicting million-byte sequences with multiscale transformers. *Advances in Neural Information Processing Systems*, 36:78808–78823, 2023.

Xiang Yue, Tianyu Zheng, Yuansheng Ni, Yubo Wang, Kai Zhang, Shengbang Tong, Yuxuan Sun, Botao Yu, Ge Zhang, Huan Sun, Yu Su, Wenhu Chen, and Graham Neubig. Mmmu-pro: A more robust multi-discipline multimodal understanding benchmark, 2024. URL `https://arxiv.org/abs/2409.02813`.

Renrui Zhang, Dongzhi Jiang, Yichi Zhang, Haokun Lin, Ziyu Guo, Pengshuo Qiu, Aojun Zhou, Pan Lu, Kai-Wei Chang, Yu Qiao, et al. Mathverse: Does your multi-modal llm truly see the diagrams in visual math problems? In *European Conference on Computer Vision*, pp. 169–186. Springer, 2025.

Ruohong Zhang, Bowen Zhang, Yanghao Li, Haotian Zhang, Zhiqing Sun, Zhe Gan, Yinfei Yang, Ruoming Pang, and Yiming Yang. Improve vision language model chain-of-thought reasoning, 2024. URL `https://arxiv.org/abs/2410.16198`.

Boyuan Zheng, Boyu Gou, Jihyung Kil, Huan Sun, and Yu Su. Gpt-4v (ision) is a generalist web agent, if grounded. In *Forty-first International Conference on Machine Learning*, 2024.

## A  Example Input

Figure 8 and Figure 9 gives two examples about the vision input.

## B  Attention Heatmap before and after ImageFast Method

Figure 10 presents a heatmap comparison between PEAP and PEAP-Fast. PEAP-Fast effectively reduces redundant patches while preserving attention on key regions.

## C  Attention Heatmap Comparison Between Datasets

We provide additional attention visualizations on three representative datasets—*MBPP*, *MMLU-Pro*, and *MathVerse*—to illustrate how attention patterns vary across program synthesis, reasoning, and STEM tasks.

You are a table analyst. Your task is to answer questions based on the table content.

The answer should follow the format below:

[Answer Format]

Final Answer: AnswerName1, AnswerName2...

Ensure the final answer format is the last output line and can only be in the "Final Answer: AnswerName1, AnswerName2..." form, no other form. Ensure the "AnswerName" is a number or entity name, as short as possible, without any explanation.

Give the final answer to the question directly without any explanation.

Read the table in the below image.

| 0 | Unnamed: 0 | airdate | episode | rating | share | rating / share (1849) | viewers (millions) | rank (timeslot) | rank (night) |
|---|---|---|---|---|---|---|---|---|---|
| 1 | 1 | february 14 , 2010 | nanna is kickin' your butt | 5.100000 | 8 | 2.8 / 7 | 9.070000 | 1 | 1 |
| 2 | 2 | february 21 , 2010 | when the cow kicked me in the head | 5.200000 | 8 | 2.9 / 7 | 9.110000 | 1 | 1 |
| 3 | 3 | february 28 , 2010 | run like scalded dogs! | 5.800000 | 9 | 3.2 / 8 | 10.240000 | 2 | 4 |
| 4 | 4 | march 7 , 2010 | we are no longer in the bible belt | 4.500000 | 7 | 2.6 / 7 | 8.050000 | 2 | 4 |
| 5 | 5 | march 14 , 2010 | i think we 're fighting the germans , right | 5.800000 | 10 | 3.0 / 9 | 10.100000 | 1 | 3 |
| 6 | 6 | march 21 , 2010 | cathy drone | 6.900000 | 11 | 3.8 / 9 | 11.990000 | 1 | 4 |
| 7 | 7 | march 28 , 2010 | anonymous | 7.200000 | 11 | 3.9 / 10 | 12.730000 | 1 | 3 |
| 8 | 8 | april 4 , 2010 | you 're like jason bourne , right | 5.200000 | 9 | 2.7 / 8 | 9.140000 | 1 | 3 |
| 9 | 9 | april 11 , 2010 | dumb did us in | 6.900000 | 11 | 3.4 / 10 | 11.880000 | 1 | 3 |
| 10 | 10 | april 25 , 2010 | i feel like i'm in , like , sicily | 6.300000 | 10 | 3.2 / 9 | 10.690000 | 1 | 3 |
| 11 | 11 | may 2 , 2010 | they don't even understand their own language | 6.000000 | 10 | 3.0 / 9 | 10.290000 | 1 | 3 |

Let's get start!

Question: How many episodes had a rating of 5.3 or higher?

Figure 9: An example input of TableBench dataset, using Direct Prompt.

## D    Broader Impact Statement

This work explores a unified pixel-based perception paradigm that eliminates the need for separate text and image tokenization. While such unification can simplify multimodal pipelines and reduce reliance on OCR systems, it also introduces new risks. The computational cost of pixel-based models remains significantly higher than text-based counterparts, which may limit accessibility and increase the carbon footprint of large-scale training and deployment. Furthermore, because pixel inputs may contain sensitive visual information, researchers must ensure that data synthesis and collection comply with privacy and ethical standards.

On the positive side, the PixelWorld benchmark provides a transparent and reproducible foundation for assessing multimodal understanding, encouraging fair comparisons across models and modalities. By highlighting where pixel-based representations succeed and fail, this work aims to guide the community toward more efficient and interpretable multimodal systems, fostering broader exploration of unified perception without compromising ethical or environmental considerations.

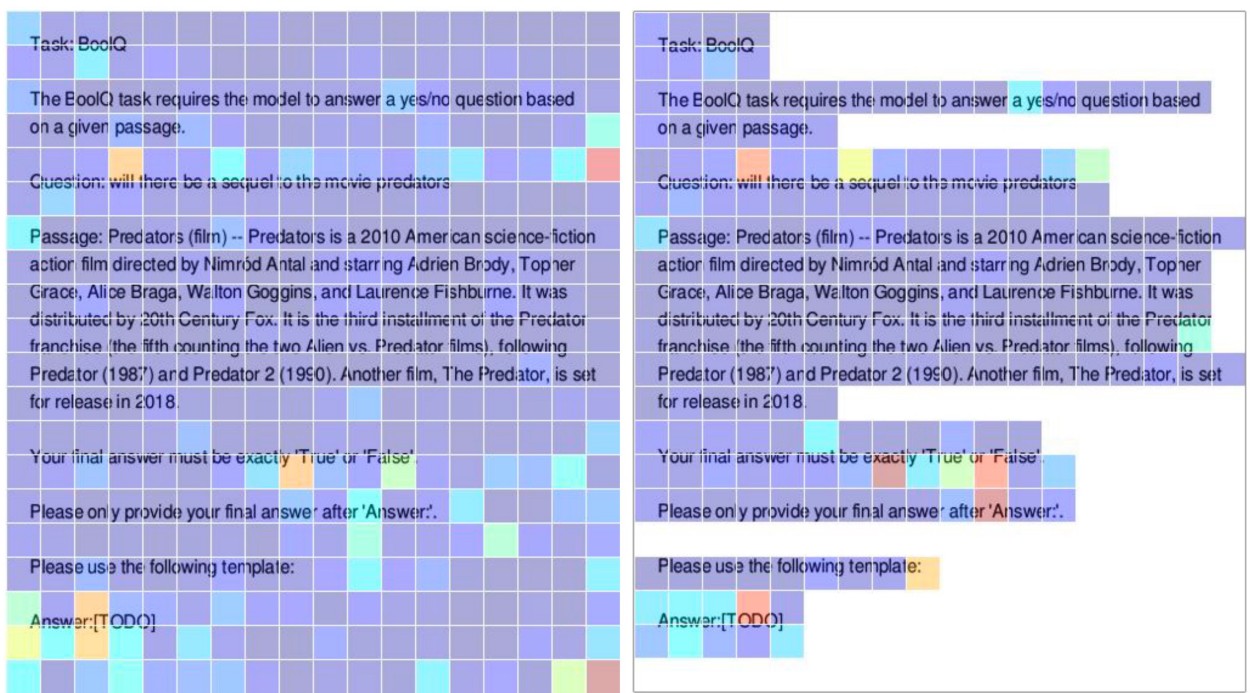

Figure 10: Last Layer Attention Heatmap on Qwen2VL-7B between PEAP (left) and PEAP-Fast (right).

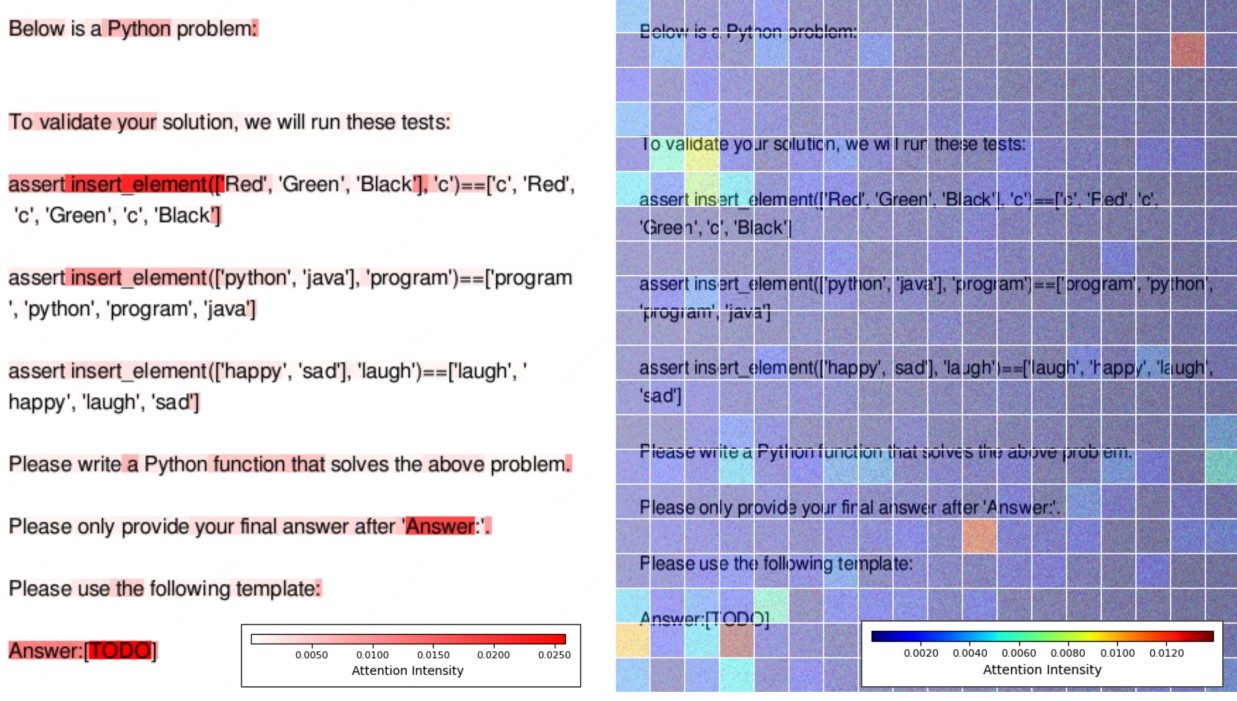

Figure 11: Example 1: Last Layer Attention Heatmap on Qwen2VL-7B from MBPP.

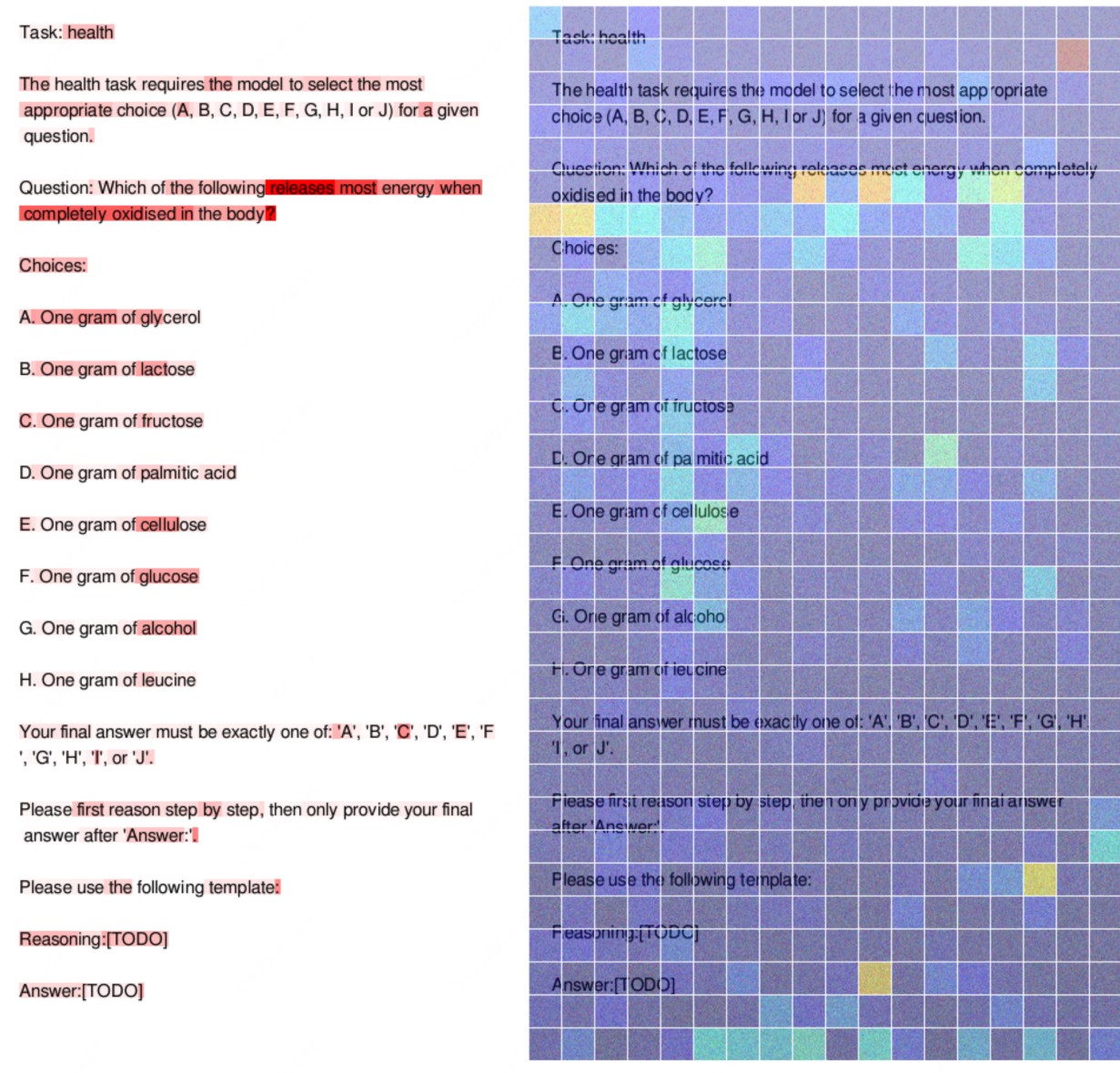

Figure 12: Example 2: Last Layer Attention Heatmap on Qwen2VL-7B from MMLU-Pro.

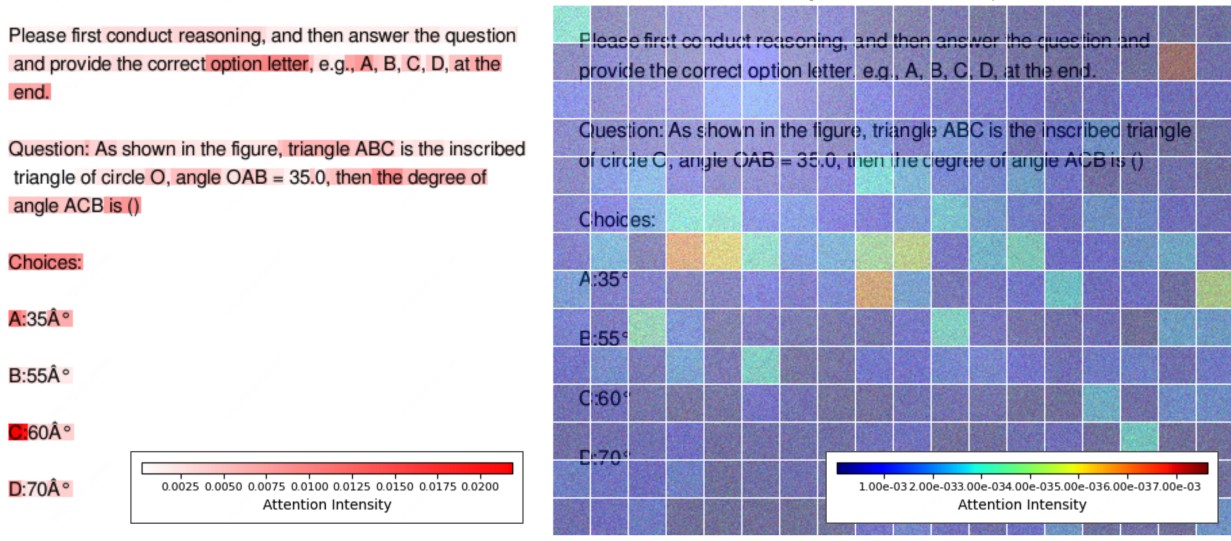

Figure 13: Example 3: Last Layer Attention Heatmap on Qwen2VL-7B from MathVerse.

