# OpenReview forum: "PixelWorld: Towards Perceiving Everything as Pixels"
_TMLR — Accepted by TMLR_

### Review · Reviewer_Vdbc · 2025-07-30

**Summary Of Contributions:**

**Summary:**

The authors propose to Perceive Everything as Pixels (PEAP) as a method to address the challenges of understanding text and raw image together, instead of treating them as different modalities and employing separate image and text models.

They argue that "modality-specific processing leads to a fragmented understanding of multimodal inputs and increases the complexity of engineering pipelines," especially when these inputs are heavily intertwined.

They also introduce a benchmark, PixelWorld, containing ten benchmarks covering different tasks and modalities. PixelWorld encodes different modalities of natural language, tabular data, code, mathematical, and diagrammatic inputs into a unified pixel space. They include both token-based (baseline) and pixel-based (their idea) formats.
The authors evaluate different vision–language models such as Qwen2VL-2B and GPT-4o on the proposed benchmark in a zero-shot manner.

They find that PEAP's performance is good for semantic understanding (especially in intrinsically multimodal settings such as website rendering), but reduces accuracy for reasoning tasks such as math and coding.

The authors conduct three additional analyses, including:
1. Representation analysis: finding consistent structures when comparing the attention maps of token-based inputs versus pixel-based inputs
2. Efficiency optimization: introducing PEAP-fast to prune empty patches, which results in an 80% speedup with the same accuracy.
3. Prompt sensitivity: Investigating different prompting methods. The authors find the chain-of-thought prompting to be the most promising.

**Strengths:**
1. Addressing an interesting and valid challenge (encoding intertwined inputs) in the multimodal learning domain.
2. Proposing an interesting solution to treat all input modalities as pixels.
3. Developing an image data synthesis for text-only datasets. This can be useful to the multimodal learning community by providing more multimodal datasets.
4. The paper is very well-written and the authors communicate their method and findings clearly.

**Weaknesses:**
1. Even though I agree that intertwined text and image should be treated as the same modality, encoding everything into pixels has the disadvantage of not leveraging the power of language models.
2. It seems to be overstating to suggest that vision encoders can act as a universal multimodal tokenizer based on the similar global patterns between pixel-based and token-based inputs. What about intertwined text and image data where the image is not synthesized from text? What about sequential characteristics present in text, but not in images?
3. How is the global attention similarity computed? Is it a subjective evaluation by the authors?
4. The computational overhead of encoding inputs into pixels, combined with the reduced or on-par performance in most cases, makes it difficult to justify the benefits of PEAP.

**Audience:**

Yes

**Audience Explanation:**

The problem that the authors address is challenging and important to the multimodal machine learning community. It would be beneficial to the TMLR's audience to learn about the proposed method and its strengths and limitations.

**Broader Impact Concerns:**

There is no Broader Impact Statement section in the paper, but there are no ethical concerns. However, the authors could include this section and mention the potential harms, biases, and risks inherent in large models.

**Claims And Evidence:**

Yes

**Claims Explanation:**

The authors do a great job conducting various and comprehensive experiments and they explain their results in a clear manner. However, the results are not completely convincing as to why the proposed method would be beneficial.

**Requested Changes:**

1. A discussion about the benefits of the proposed method given the extra computations necessary for this method and not enough performance gains in return.
2. Figure 1 is dense and should be broken into at least two different figures, one for the method and one for insights.
3. Numbers on Figure 1 need explanation in the caption or on the axis of the charts.
4. Figure 1, why does the input go to ViT and then the Language Model? In the body of the paper, it seems like the input (after being mapped into pixel space) goes to a VLM.

---

> ### Author Response · Authors · 2025-08-19
>
> ## On Encoding Everything into Pixels and the Power of Language Models
>
> We agree that encoding all inputs into pixels may lose some of the specialized strengths of language models. Our claim that vision encoders can act as universal multimodal tokenizers was not intended to overstate current capabilities. Rather, our observation of similar global attention patterns between pixel-based and token-based inputs suggests that certain language modeling abilities can generalize into the image space. This is an empirical insight, not a claim that vision encoders already match language models in all aspects.
>
> Regarding sequential structures, we argue that spatial positions in images can also encode order information, sometimes even more explicitly by separating unrelated text into different regions. Importantly, our evaluation goes beyond synthetic settings: on naturally sampled tasks such as **SlidesVQA** and **Wiki-SS**, PEAP outperformed OCR-based text baselines. These results highlight the advantages of pixel-based representation in **computer-use scenarios where text and images are inherently intertwined**, such as GUI agents and document understanding.
>
> To clarify, we are not advocating that PEAP should replace text tokenization for purely textual tasks, where language models clearly have an advantage. Instead, we position PEAP as a **more universal solution for interwoven multimodal inputs**, where maintaining separate tokenizers can introduce complexity and overhead. In such real-world scenarios, the universality of pixel encoding justifies its computational cost, even if it is not the optimal choice for text-only tasks. For broader real-world physical data, we agree this remains an important direction for future work.
>
>
> ## Formula for Global Attention
>  The attention heatmaps were computed during greedy decoding by averaging the last-layer attention across tokens. For multi-head attention, we applied simple averaging. The formula is:
>
> $$
> \text{Heatmap}(i) \;=\; \frac{1}{H} \sum_{h=1}^{H} \big| A^{(L)}_{h}[t, i] \big|,
> \quad i \in [s, e)
> $$
>
> where:
> - $H$ = number of attention heads
> - $L$ = index of the last layer
> - $A^{(L)}_{h}[t, i]$ = attention weight from token \(t\) to token \(i\) in head \(h\)
> - $[s, e)$ = decoding range during greedy decoding
>
> ## On the Benefits of PEAP Given Computational Overhead
>
>  Our primary emphasis is on the generality of PEAP. In GUI agent scenarios, for instance, text and images are inherently intertwined, and models must inevitably extract textual information from pixels. This is fundamentally different from reasoning directly over clean text inputs. PixelWorld was designed to highlight these differences and to call attention to the performance gap that emerges. We also acknowledge that for pure-text tasks, PEAP may not be the most appropriate approach.
>
>
> ## Figure 1 Layout and Clarity
>
>  We appreciate the suggestion and will revise Figure 1 by simplifying the content, breaking it into two separate figures (one for the method and one for insights), and adding explanations for the numbers either in captions or axes. The detailed contents of Figure 1 are further elaborated in later figures, and we will make this linkage more explicit.
>
>
> ## Clarification of Model Architecture in Figure 1
>  Taking Qwen2.5-VL as an example, it consists of a visual encoder (ViT) and a language decoder. In Figure 1, our intention was to illustrate how tokenization occurs within a VLM, but we inadvertently underemphasized the overall architecture. In the revised version, we will use a dashed box to clearly indicate that both ViT and the language decoder are integral components of the VLM.
>
> ## Broader Impact Session
> Thanks for the suggestion — we will adding a Broader Impact Statement in the final revision.

---

> > ### Comment · Reviewer_Vdbc · 2025-09-01
> >
> > Thank you for the response and clarifications.

---

### Review · Reviewer_hHmm · 2025-08-04

**Summary Of Contributions:**

The paper proposes a unified perception paradigm as Perceive Everything as Pixels (PEAP) to perceive text, images, code as pixel-based visual data, instead of custom discrete tokenization. Further, the work proposes PEAP-Fast, an algorithm that prunes blank patches and achieves up to 80% speedup without accuracy loss compared to naive PEAP. The paper explores attention patterns of Qwen2VL-7B and  prompt sensitivity, especially CoT for reasoning tasks.

**Audience:**

Yes

**Audience Explanation:**

The work provides interesting analysis on diverse kinds of tasks that can be considered relevant for pixel-only based processing, and how the current SOTA models compare among these benchmarks.

**Broader Impact Concerns:**

The work proposes perceiving everything as pixels-only, which seems promising but might have broader concerns about vulnerability of pixel-only approach, compute overhead, latency, etc.

**Claims And Evidence:**

No

**Claims Explanation:**

“Figure 6: Last Layer Attention Heatmap on Qwen2VL-7B between token-based (left) and pixel-based (right)” - what is the colorbar for this? How to interpret the attention colors on the right? This will also clarify "Figure 9: Last Layer Attention Heatmap on Qwen2VL-7B between PEAP (left) and PEAP-Fast (right)."

Why ROUGE-L only as a metric for QA tasks? Why not consider semantic similarity based metrics too, as ROUGE-L will check for exact match only. Any interesting failure case analysis for these tasks should be highlighted.

How do models trained for reasoning (like o-series, deepseek-r1, Gemini-2.5-pro, etc.) perform on these tasks?

**Requested Changes:**

Please proofread the work and fix missing details as the paper seems to be written with AI assistance.
For eg.,
- what version of Gemini Flash is used? 2.0 or 2.5?
- what is the colorbar for attention masks?
- In section 4.3 Q3, What is implied by "massive experimental results"? I see both kinds of tasks where "text" is better than "img" input and vice versa. Explain this with specificity and cross references.

---

> ### Author Response · Authors · 2025-08-19
>
> We thank the reviewer for the insightful comments and suggestions. We provide clarifications and planned improvements below.
>
> ## Attention Heatmap Colorbar and Interpretation
>
>  We normalized attention values to the range [0, 1] using a_norm = (a - min) / (max - min). For text attention heatmaps, darker red indicates stronger attention. For image attention, we applied OpenCV’s `COLORMAP_JET`, where brighter colors represent higher saliency. The raw attention ranges were [1.3e-5, 2.4e-2] for text and [0.0, 9.1e-3] for images. In the revision, we will add legends and clarify these ranges to make the figures interpretable.
>
> ## Evaluation Metrics for QA Tasks
>
>  ROUGE-L was used primarily as a practical baseline because of its wide adoption in QA evaluation. We acknowledge that semantic similarity–based metrics such as BERTScore or LLM-based judges can capture meaning beyond surface matching, while we expect these would not alter our core conclusions.
>
> ## Reasoning-Trained Models
>
>  We attempted to evaluate Gemini-1.5-pro but encountered usage limitations and therefore did not report the results. At the time of writing (early 2025), advanced reasoning models such as OpenAI o1 and Deepseek R1 did not support image inputs, preventing direct multimodal comparison. We agree that benchmarking reasoning-trained models will provide valuable perspective, and plan to extend PixelWorld evaluations as such support becomes available.
>
> ## Gemini Flash Version
>
>  We confirm that the version used was gemini-1.5-flash-002 and will make this explicit in the final version.
>
> ## Clarity of Section 4.3 (Q3)
>
>  Our intention in Section 4.3 was to show that in purely textual synthesis tasks, image-based inputs consistently underperform text inputs. We then analyzed whether chain-of-thought methods could reduce this gap. We will revise the text to state this clearly, replace vague expressions such as “massive experimental results” with dataset-specific references, and provide cross-links to Section 3.1 for consistency.

---

### Review · Reviewer_P6GJ · 2025-08-05

**Summary Of Contributions:**

### Summary
This work aims to stress-test the existing Vision-Language Models (VLMs) in perceiving inputs as pixels, using 10 different benchmark datasets that cover text-only, structured table-based, and multimodal scenarios. The authors then show that VLMs (when processing inputs including text as pixels, compared to processing them as pure text) obtain competitive performance on relatively simple semantic understanding tasks (such as data analysis or fact checking); however, on reasoning-intensive tasks (such as math or code), VLMs show substantial performance drops, especially when they are small. Lastly, to mitigate the issue of high latency in processing inputs as pixels, the authors propose pruning black patches, on which there is no meaningful information for text or images, and show that this simple strategy effectively accelerates the inference speed with a marginal accuracy drop.

---

### Strengths
* While the idea of perceiving every input as pixels is not new (and has been explored), the authors provide an extensive study on it.
* The proposed pruning strategy for black patches is simple yet effective, offering a practical way to reduce latency with minimal impact on performance.

---

### Weaknesses
* The idea of perceiving every input as pixels is not new, and the findings provided by this paper are also similar to existing work. For example,  Faysse et al. (2024) and Ma et al. (2024), discussed in the related work section, already show promise in handling multimodal inputs (consisting of text and images) as pixels, and the performance degradation of VLMs on complex reasoning tasks is well known in the field [A, B]. In this regard, it might be questionable what additional benefits this work offers over them.
* The paper provides the experimental results, but does not provide the deep intuition on why VLMs are still ineffective in certain scenarios. For example, in reasoning or more complex tasks, why are there significant performance drops when VLMs are processing inputs as pixels, which are especially more severe when model sizes are small, and similarly, why are there more challenges when VLMs are processing mixed-modality inputs? I think, beyond numbers from the experimental figures or tables, the readers are more interested in the rationales related to them.
* In Section 4.1, the authors claim that VLMs behave similarly on textual and image inputs with one single example with one single VLM, which I feel is insufficient. It would be necessary to provide more examples and their aggregated scores with multiple VLMs to demonstrate this claim.
* The title of this paper may not be aligned with the content of this paper. To my understanding, this paper is more like a VLM benchmark paper (with the benchmark named PixelWorld), but when looking at its title, I do not feel that. I think it may be better to provide a little more detail (or nuance) on the paper title.

---

[A] Improve Vision Language Model Chain-of-thought Reasoning, 2024

[B] Beyond Captioning: Task-Specific Prompting for Improved VLM Performance in Mathematical Reasoning, 2024

**Audience:**

No

**Audience Explanation:**

VLMs are a topic of interest. However, the way to process inputs as pixels is already known, and most of the findings provided by this paper (such as the high performance on relatively simple semantic understanding tasks and the low performance on difficult reasoning tasks) are already well-known and thus not particularly novel.

**Broader Impact Concerns:**

The authors do not provide the Broader Impact Statement section. However, I do not see any major concerns about it.

**Claims And Evidence:**

Yes

**Claims Explanation:**

If the goal of this paper is to benchmark VLMs in perceiving inputs as pixels, I think it is supported by multiple experimental results with 10 different benchmark datasets. However, the findings provided by this paper are not entirely new and are similar to those from existing works. Also, this paper does not provide deep intuition on why VLMs are suboptimal on some benchmark results. Lastly, the claim made in Section 4.1 might be specific to a single case and should be backed by more experimental results.

**Requested Changes:**

Please see my comments in the Weaknesses section above.

---

> ### Author Response · Authors · 2025-08-19
>
> We thank the reviewer for their thoughtful comments and constructive feedback. Below we clarify our contributions and address the concerns raised.
> ## Novelty and Contributions
> While prior works such as Faysse et al. (2024) and Ma et al. (2024) have indeed explored the idea of treating inputs as pixels, their core focus is on demonstrating that VLMs are weak at reasoning and proposing methods (e.g., long-context training, prompting strategies) to alleviate this limitation. In contrast, our work pushes this line of research forward in two distinct ways:
>
> 1. We introduce **PixelWorld**, a benchmark that enables a *quantitative and explicit comparison between visual and textual tokenizers under a unified framework*. Unlike prior studies that only explored the reasoning performance of pixel-based approaches, PixelWorld provides a systematic and measurable basis for evaluating modality trade-offs, which is essential for guiding future development.
>
> 2. Based on insights from PixelWorld, we propose **PEAP-Fast**, an inference-acceleration method that significantly improves the efficiency of pixel-based VLMs with only minor accuracy degradation. This represents a practical engineering contribution that complements prior efforts.
>
> Together, these contributions move beyond simply observing the limitations of VLM reasoning and instead establish both a benchmark for fair comparison and a solution for advancing the use of pixel-based tokenization in applications such as GUI agents, where text and images are inherently intertwined.
>
> ## Lack of Intuitive Explanations
>  We appreciate this suggestion. In Section 4.1, we used attention pattern analysis to show that the image tokenizer plays a similar role to the text tokenizer. This provides an intuitive understanding: pixelized inputs preserve consistency in lower-level semantic tasks but struggle in higher-level reasoning tasks, where stronger cross-modal generalization is required.
>
> ## Attention Visualization with More Examples
>  This is an excellent point. In fact, we have observed similar attention patterns across multiple datasets. For space reasons, SuperGLUE was presented as a representative example in the main text. We will add more examples and aggregated statistics in the appendix of the revised version. A preliminary collection of additional cases is available [here](https://docs.google.com/document/d/1Ggv305ZGwpxhE5MrAzDdtkDTk9edgcnIl1P8vPrpgDQ/edit?tab=t.0).
>
>
> ## Paper Title
>  Thanks for the suggestion. We will revise the title to better emphasize the benchmarking aspect, while still highlighting the pixel-based unification perspective.

---

> > ### Comment · Reviewer_P6GJ · 2025-08-26
> >
> > Thank you for your response. It mitigates some of my concerns and questions, specifically regarding the points about the contribution and more visualization examples. On the other hand, it would be great if the authors could provide more intuitive explanations and analyses (i.e., the core reasons behind the ineffectiveness of VLMs) beyond simply saying VLMs are still ineffective in certain scenarios, as well as providing a more specific answer to the future paper title.

---

> ### Author Response · Authors · 2025-08-28
>
> We sincerely thank the reviewer for the follow-up comments and for recognizing the clarifications provided in our earlier response.
>
> Regarding the paper title, we agree with the suggestion to make it more specific and aligned with the benchmarking nature of the work. In the revised version, we plan to adopt the title: *"PixelWorld: How Far Are We from Perceiving Everything as Pixels?"*. We believe this reflects both the benchmarking aspect and the unifying perspective of treating all inputs as pixels.
>
> On the request for deeper intuitive explanations, we agree this is an important direction. However, the scope of this work is to evaluate and benchmark the (in)effectiveness of pixel-based VLMs. Explaining the precise mechanisms behind these limitations requires further theoretical and empirical study, which we see as a natural direction for future work.

---

### Author Response · Authors · 2025-09-01
**Overall Comment**

We thank the reviewers for their thoughtful assessments.

Strengths were consistently recognized: PixelWorld as a comprehensive and relevant benchmark for tightly intertwined text–image scenarios (P6GJ, Vdbc), clear communication of findings (Vdbc), and PEAP-Fast as a practical latency reducer with minimal accuracy impact (P6GJ, hHmm).

The main shared concern is novelty/positioning relative to prior “pixels-only” works, along with requests for deeper intuition, clearer visualizations/metrics, and discussion of compute overhead. In our rebuttal, we clarified scope and contribution: this work is explicitly benchmark + engineering—(i) a unified, fair comparison of pixel vs. token representations and (ii) a deployable efficiency knob (PEAP-Fast)—and we explained why this matters for GUI/document use cases where pixels are unavoidable. We also resolved misunderstandings by (a) detailing attention-heatmap normalization/interpretation and noting broader examples we can aggregate, (b) explaining the choice of ROUGE-L and (c) documenting multimodal constraints for reasoning-centric models (o-series, R1, Gemini) that limited head-to-head comparisons. We will incorporate these clarifications (including a more precise title emphasizing benchmarking) in the revision.

---

### Decision · Action_Editor_gP94 · 2025-09-09

**Recommendation:** Accept with minor revision

**Additional Comments:**

The reviewers reached mixed recommendations: one "Leaning Reject," one "Leaning Accept," and one "Leaning Accept." The primary concerns centered on limited novelty (extending known paradigms rather than introducing fundamentally new approaches) and insufficient depth in explaining performance differences. However, reviewers consistently recognized the benchmark's utility and PEAP-Fast's practical value. The authors effectively addressed methodological clarifications and positioning concerns in rebuttals, emphasizing their contribution as systematic benchmarking rather than algorithmic innovation.

The paper provides valuable evaluation and practical insights that merit publication, contingent on addressing the clarifications promised in rebuttals. The authors should incorporate the methodological details, attention heatmap legends, additional visualization examples, and clearer positioning of the benchmarking contribution as discussed in their responses.

**Audience:**

Yes

**Audience Explanation:**

The TMLR audience would find value in this systematic benchmarking study. The work addresses practically relevant scenarios where text and images are inherently intertwined (GUI agents, document processing), and provides a unified evaluation framework for comparing modality representations. While the core finding that pixel-based approaches underperform on reasoning tasks aligns with prior knowledge, the comprehensive benchmark and efficiency optimization (PEAP-Fast) offer practical utility to the multimodal learning community.

**Claims And Evidence:**

Yes

**Claims Explanation:**

The paper's core claims about benchmarking pixel-based vs. token-based processing are supported by experimental evidence across 10 datasets. However, several evidence gaps were identified: (1) insufficient examples to support claims about attention pattern similarities between pixel and token processing (only one example provided initially), (2) unclear attention heatmap interpretation and normalization methods, and (3) limited evaluation of state-of-the-art reasoning models due to multimodal constraints. The authors addressed some concerns in rebuttals by clarifying methodological details and providing additional examples, but reviewers noted the need for deeper intuitive explanations beyond empirical observations.